# Comparative Evaluations on Real-Time Monitoring of Temperature Sensors during Endoscopic Laser Application

**DOI:** 10.3390/s23136069

**Published:** 2023-06-30

**Authors:** Minh Duc Ta, Van Gia Truong, Seonghee Lim, Byeong-Il Lee, Hyun Wook Kang

**Affiliations:** 1Industry 4.0 Convergence Bionics Engineering, Marine-Integrated Biomedical Technology Center, Pukyong National University, Busan 48513, Republic of Korea; tmduc1997@pukyong.ac.kr (M.D.T.); seonghee@pukyong.ac.kr (S.L.); 2TeCure, Inc., Busan 48548, Republic of Korea; gvtruong@tecure.co.kr; 3Division of Smart Healthcare and Digital Healthcare Research Center, College of Information Technology and Convergence, Pukyong National University, Busan 48513, Republic of Korea; 4Marine-Integrated Biomedical Technology Center, The National Key Research Institutes in Universities, Pukyong National University, Busan 48513, Republic of Korea

**Keywords:** Fiber Bragg Grating, diffusing fiber, thermal coagulation, thermocouple

## Abstract

Temperature sensors, such as Fiber Bragg Grating (FBG) and thermocouple (TC), have been widely used for monitoring the interstitial tissue temperature during laser irradiation. The aim of the current study was to compare the performance of both FBG and TC in real-time temperature monitoring during endoscopic and circumferential laser treatment on tubular tissue structure. A 600-µm core-diameter diffusing applicator was employed to deliver 980-nm laser light (30 W for 90 s) circumferentially for quantitative evaluation. The tip of the TC was covered with a white tube (W-TC) in order to prevent direct light absorption and to minimize temperature overestimation. The temperature measurements in air demonstrated that the measurement difference in the temperature elevations was around 3.5 °C between FBG and W-TC. Ex vivo porcine liver tests confirmed that the measurement difference became lower (less than 1 °C). Ex vivo porcine esophageal tissue using a balloon-integrated catheter exhibited that both FBG and W-TC consistently showed a comparable trend of temperature measurements during laser irradiation (~2 °C). The current study demonstrated that the white tube-covered TC could be a feasible sensor to monitor interstitial tissue temperature with minimal overestimation during endoscopic laser irradiation. Further in vivo studies on gastroesophageal reflux disease will investigate the performance of the W-TC to monitor the temperature of the esophageal mucosa surface in real-time mode to warrant the safety of endoscopic laser treatment.

## 1. Introduction

The esophagus is a muscular tubular tissue that serves as the beginning of the digestive tract. Its main function is to transport food from the larynx to the stomach [1]. The esophagus is located in the middle of the chest of a human and has an average length of 18~25 mm with a normal diameter of 20~30 mm [2,3,4,5,6]. Esophageal disorders are a group of conditions that affect the function of the esophagus, including infectious esophagitis, drug-induced esophagitis, radiation-induced esophagitis, caustic esophagitis, and esophageal cancers [7]. Patients typically suffer from a variety of symptoms, such as difficulty swallowing, chest pain, acid reflux, sore throat, and unintentional weight loss. Among a number of treatments, endoscopic therapy is considered a minimally invasive procedure that uses an endoscope (a flexible and lighted tube) to treat a range of conditions affecting the esophagus. Common endoscopic therapies for esophageal diseases include endoscopic dilation (PD), endoscopic mucosal resection (EMR), endoscopic submucosal dissection (ESD), and radiofrequency balloon ablation [8,9,10,11]. In particular, a balloon catheter-integrated diffusing applicator (BCDA) has recently been developed as a potential therapeutic device for treating tubular tissue structures effectively [12,13,14]. The combined treatment with balloon dilation and cylindrical thermal coagulation can expand the tubular tissue, selectively denature the abnormal tissue, and remodel the collagen in the tubular structure. Both ex vivo and in vivo esophageal tissue experiments have demonstrated that BCDA-combined endoscopic procedures showed promising results in terms of selective tissue ablation/coagulation and minimal thermal injury to the adjacent tissue [13,15,16]. However, laser absorption generates heat in the tissue, and excessive thermal energy can cause damage to the surrounding healthy tissue. Real-time temperature monitoring can allow physicians to closely monitor tissue temperature during the procedure, ensuring safety margins and minimizing the risk of thermal injury [17]. In particular, during endoscopic laser treatment, the temperature of the esophageal mucosa is maintained below 50 °C to preserve the mucosa surface [15,16]. Thus, real-time temperature monitoring can play a critical role in ensuring the safety, precision, and effectiveness of endoscopic laser treatments, allowing physicians to optimize laser dosimetry while minimizing the risk of complications.

Two techniques are available for real-time temperature measurement: non-invasive and invasive methods. Non-invasive methods include infrared thermography, ultrasound, magnetic resonance imaging (MRI), and laser opto-acoustics (OA) [18]. However, these methods have limitations, such as lack of accuracy and long acquisition time, and they are not suitable for clinical settings. On the other hand, invasive techniques consist of various commercial sensors, such as Fiber Bragg Grating (FBG), thermocouple (TC), and optical sensors. FBG sensors have several advantages, including small physical dimensions, light weight, high optical sensitivity, wide working bandwidth, biocompatibility, and chemical inertness [19,20]. Therefore, FBG sensors are suitable for various medical purposes that require high resolution, in-body insertion, fabric integration, tissue biomechanics, and medical bio-sensing, due to their small size, good metrological properties, and intrinsic biocompatibility [21]. However, FBG sensors are still expensive and have low tensile strength because of the pure silica (SiO_2_) material [22,23]. TCs are considered to be a lower-cost alternative to FBG sensors that can achieve the same performance. The TCs are relatively accurate (around 1 °C), respond quickly to temperature changes (within a few hundred milliseconds), and are immune to shock and vibration [24,25,26]. However, the main drawback of TCs is light absorption due to the presence of metal, resulting in self-heating and overestimation during real-time temperature monitoring in laser treatment [24]. The current study aimed to compare the performance of FBG and TC covered with a white plastic tube to minimize self-heating and overestimation errors during endoscopic laser treatment of tubular tissue. It was hypothesized that the protection of the TC tip could prevent light absorption during irradiation and reduce the measurement differences between FBG and TC.

## 2. Materials and Methods

### 2.1. Light Delivery

A diffusing applicator (DA) with a 600 μm core diameter (TeCure, Inc., Busan, Republic of Korea) was used to transmit cylindrical laser light along a 10-mm active tip. The diffusing tip was fabricated using a CO_2_ laser micro-machining system [27]. After the fabrication, the tip was sealed with epoxy with a customized glass cap (5 mm in length and 1.4 mm in outer diameter; OD) for mechanical protection. To evaluate the spatial light distribution from the fabricated DA, a customized goniometric system, in conjunction with a photodiode (PD-300-3W, Ophir, Juresalem, Isreal), was used to measure longitudinal and polar emissions [27]. To validate the goniometric results, a longitudinal emission from the DA tip was captured using a digital camera (D5100, Nikon Corp., Tokyo, Japan) and analyzed by Image J software (Version 1.53t, National Institute of the Health, Bethesda, MD, USA) for quantitative comparison. He-Ne laser light (λ = 632 nm, Thorlabs Inc., Newton, NJ, USA) was coupled into the DA for all emission measurements. In Figure 1a, a He-Ne image clearly shows a uniform light distribution along the DA tip. Figure 1b shows longitudinal emission profiles measured by both the goniometric system (blue line) and digital imaging (red line). The longitudinal emissions exhibited a profile of two symmetrical peaks in the light distribution (2.5 and 7.5 mm positions). The drop in intensity occurred in the middle of the DA tip and yielded a lower intensity (≤25%), compared to the highest peak. Overall, the shape of the profiles was similar for both measurements. According to Figure 1c, the normalized polar intensities ranged from 0.83 to 1.00 with a deviation of less than 10%, indicating homogeneous and cylindrical light emission from the DA.

After performance validation tests, a BCDA was assembled using the fabricated DA and a customized transparent balloon. The length and diameter of the balloon were 30 mm and 22 mm, respectively. The DA tip was positioned in the center of the balloon and sealed with epoxy on both sides of the balloon. The balloon was inflated with a cold filling medium (water) using an inflator. To minimize the amount of air trapped inside the balloon, the balloons were repetitively deflated using the inflator [28]. A 980 nm diode laser system (EsoLight Z360, TeCure Inc., Busan, Republic of Korea) was used for the current study as a laser source. Settings of 30 W and 90 s were chosen for the applied power and irradiation time, respectively.

### 2.2. Temperature Sensors

A FBG sensor (T-1-N-GL-FI, Micronor Sensors LLC, Sperry Ave, Ventura, CA, USA) was used as a reference in the current study. The measurement precision of the FBG was 0.1 °C and stable for up to 350 °C. The measurement principle of the FBG is based on the Bragg wavelength shift phenomenon caused by the induced temperature. By tracking the reflected spectrum of the FBG, the temperature changes along the FBG structure can be measured. The reflection spectra of the FBG were monitored by a single-channel FiSpec interrogator (FBGX100, Micronor Sensors LLC, Sperry Ave, Ventura, CA, USA). The data from the interrogator were displayed in real-time on a program provided by the manufacturer. A type K TC sensor (5SRTC-TT-K-36-72, Seongnam, Gyeonggi, South Korea) was tested for real-time temperature monitoring during laser irradiation (accuracy of ±1.2 °C). A white Pebax tube (inner diameter = 0.43 mm and thickness = 0.13 mm) was used to protect the TC tip to minimize self-heating and overestimation during irradiation. The TC tip (0.25 mm in OD) was inserted into the plastic tube and fixed by epoxy in an attempt to prepare the white-tube covered temperature sensor (W-TC). The W-TC was recorded using a USB data acquisition system DAQ (OM-DAQ-USB-2400, OMEGA Inc., Norwalk, CT, USA). The temporal temperature variations of both FBG and W-TC were recorded synchronously at a sampling rate of 10 Hz.

### 2.3. Evaluation of Thin Plastic Tubes and Sensor Positions

A cylindrical glass tube with an OD of 18 mm (length = 80 mm and thickness = 1 mm) was first tested to emulate mechanical inflation in the tissue. The 18-mm glass tube was fixed vertically with a clamp holder. A DA was initially placed in the center and middle of the glass tube (Figure 2a). The jacket of the DA was marked with M_0_, M_2_, M_4_, and M_6_, indicating the positions of the FBG and W-TC sensors at 0, 2, 4, and 6 mm, respectively (Figure 1a and Figure 2a). M_0_ (0 mm) indicates the middle of the DA. The DA was attached to a rubber cap, which allowed the sensor position to be changed by moving upward/downward along the marked line on the DA jacket. Both the FBG and W-TC were placed in full contact with the outer surface of the glass tube to prevent any vibration during the tests (Figure 2a). Temperature measurements were performed at four sensor positions (room temperature), and each position was tested six times (N = 6).

### 2.4. Ex Vivo Liver Tissue Test

A W-TC positioned at the middle of a DA tip (M_0_) was selected for ex vivo porcine liver tissue testing. Two different glass tubes (18 mm and 22 mm in OD) were tested to evaluate the effect of tube size on temperature measurements (top; Figure 2b). The porcine liver tissue was obtained from a local slaughterhouse and stored at 4 °C to minimize any dehydration and structural deformation. The liver samples were cut into 30 × 30 × 30 mm^3^ and kept at a room temperature of 20 °C. Both 18-mm and 22-mm diameter hollow punches were used to create 18-mm and 22-mm OD holes in the tissue (20 mm deep), respectively. The glass tubes were carefully inserted into the perforated tissue to fully cover all of the working areas of the DA and temperature sensors. In addition, the same experiments were carried out with the liver tissue using a BDCA instead of the glass tubes to validate the performance of both sensors (bottom; Figure 2b). The balloon placed in the liver tissue was inflated with a cold filling medium using an inflator. The applied power level was 30 W, and a total energy of 2700 J was delivered in a continuous wave mode into the ex vivo liver tissue. Each ex vivo experiment was repeated six times (N = 6).

### 2.5. Ex Vivo Esophagus Tissue Test

Figure 2c illustrates an experimental setup for laser experiments on ex vivo porcine esophagus tissue using a BCDA. The porcine esophageal tissues were collected from a local slaughterhouse. The esophageal tissue was then cut into small pieces 60 mm in length. The BCDA was inserted into the tissue, and an inflator was used to inject a cold filling medium into the balloon at a pressure of 2~3 atm. A thermal IR camera (FLIR A325, Teledyne FLIR LLC, Wilsonville, OR, USA) was placed 40 cm above the esophagus tissue to record temperature changes on the tissue surface during laser irradiation of 30 W for 90 s. Both the FBG and W-TC sensors were attached to the balloon surface at the middle of the DA tip (M_0_ mm) in an attempt to monitor the real-time temperature of the inner esophageal surface. The initial temperature was maintained at 20 °C, and the ex vivo esophagus tests were repeated three times (N = 3).

### 2.6. Data and Statistical Analysis

All data were plotted using Origin 2019b software (OriginLab Corp., Northampton, MA, USA). The same initial setup conditions were maintained for all experiments, and temperature elevations (ΔT) were calculated to minimize any measurement errors. To compare the two temperature sensors, the measurement errors (ΔT_W-TC_ − ΔT_FBG_) were estimated as the temperature difference between the reference FBG sensor (ΔT_FBG_) and the W-TC (ΔT_W-TC_) in one period of irradiation time (90 s). The data were described as the mean ± standard deviation. Each dataset expressed the mean of the results of six independent experiments. The statistical package for social sciences (SPSS) version 20 program (SPSS Inc., Chicago, IL, USA) was used for the analysis. Mann–Whitney U tests were performed to evaluate the statistical significance of the difference between the two groups. A *p* < 0.05 was considered for statistical significance.

## 3. Results

Figure 3a shows temperature variations as a function of irradiation time measured at different sensor positions on the glass tube surface (i.e., 0, 2, 4, and 6 mm from the center of DA, denoting M_0_, M_2_, M_4_, and M_6_ positions, respectively) during irradiation of 30 W for 90 s without tissue. Irrespective of the sensor position, both FBG and W-TC confirmed that the temperature increased with irradiation time, and the temperature measured by the W-TC was slightly higher than that of the FBG at all sensor positions. Figure 3b presents the measurement difference (i.e., ΔT_W-TC_ − ΔT_FBG_) between FBG and W-TC at the different sensor positions to evaluate any measurement errors. Overall, both the tendency and extent of the measurement differences were comparable at all positions. As the sensor position moved further away from the center of the DA, the measurement difference slightly decreased. The W-TC at M_0_ measured up to 17% higher temperatures than M_6_ after the 90-s irradiation (i.e., 6.3 ± 0.8 °C for M_0_ and 5.4 ± 0.4 °C for M_6_). The largest measurement difference occurred at M_0_ (i.e., 3.7 ± 0.7 °C) while the positions of M_2_, M_4_ and M_6_ yielded measurement differences of 3.5 ± 0.2 °C, 3.5 ± 0.1 °C, and 3.2 ± 0.3 °C, respectively. It was noted that the sensor positions within the active length of the DA had an insignificant effect on the measurement error (less than 15%), indicating that the laser light emitted from the DA was evenly distributed over the surface of the glass tube.

Figure 4a compares the temporal developments of temperature in ex vivo porcine liver tissue during 30-W 980-nm laser irradiation for 90 s at the tissue-glass cap interface using two different glass tubes (18 and 22 mm in OD). Both FBG and W-TC showed that the temperature rise increased linearly with time, regardless of the glass tube. Due to the distance between the sensors and the DA, the 22-mm glass tube experienced a lower temperature rise than the 18-mm tube (*p* < 0.01). It was noted that the measurement differences between FBG and W-TC were smaller for the 22-mm glass tube, compared to the 18-mm glass tube (i.e., 13.7 ± 0.5 °C for W-TC and 12.2 ± 0.3 °C for FBG after 90-s irradiation; *p* < 0.01). Figure 4b confirms that the 22-mm glass tube accompanied smaller measurement differences (ΔT_W-TC_ − ΔT_FBG_) with smaller deviations than the 18-mm tube.

Figure 5 depicts the temperature measurements after BCDA-assisted laser irradiation in ex vivo porcine liver tissue. After the 90-s irradiation, W-TC measured up to a 12% higher temperature than FBG (5.9 ± 0.4 °C for FBG vs. 6.6 ± 0.2 °C for W-TC; *p* = 0.34; Figure 5a). According to Figure 5b, the measurement differences between FBG and W-TC were maintained below 1.1 ± 0.2 °C during irradiation, which was lower than the differences measured with glass tubes (Figure 3 and Figure 4).

To assess the feasibility of FBG and W-TC sensors for real-time temperature monitoring during laser treatment of tubular tissue, ex vivo porcine esophageal tissue was tested with a 30-W 980 nm laser light for 90 s using a BCDA (Figure 6). According to the IR images (Figure 6a), the surface temperature was distributed in a cylindrical manner along the esophageal tissue. The tissue temperature increased with the irradiation time and reached the maximum temperature after the 90-s irradiation. Figure 6b compares the temperature increases at the tissue lumen measured by FBG and W-TC and at the tissue surface measured by an IR camera. The tissue surface showed a faster and higher temperature increase than the lumen. The overall temperature increases in the lumen showed the same tendency for both sensors. After 90-s irradiation, the maximum temperature measured by the W-TC was slightly higher than that of the FBG (28.3 ± 0.2 °C for FBG vs. 30.0 ± 0.1 °C for W-TC). The temperature difference between the two sensors was merely 2.1 ± 0.4 °C, and the estimated deviation of the temperature measurement was 6.1% (Figure 6c), indicating that both FBG and W-TC had comparable measurement performance for the endoscopic laser treatment of the tubular tissue.

## 4. Discussion

The goal of this study was to compare the performance of FBG and W-TC sensors for real-time temperature monitoring during endoscopic and circumferential laser applications. The current findings demonstrated that the use of a white tube was able to minimize measurement differences (ΔT_W-TC_ − ΔT_FBG_), compared to the FBG. It was noted that the position of the sensor tip (longitudinal direction in Figure 3 and radial direction in Figure 4) could determine the extent of the measurement difference. According to Figure 3b, (ΔT_W-TC_ − ΔT_FBG_) remained almost invariant (≤15%) at various positions (i.e., 3.7 ± 0.7 °C for M_0_, 3.5 ± 0.2 °C for M_2_, 3.5 ± 0.1 °C for M_4_, and 3.2 ± 0.3 °C for M_6_). The uniform light distribution from the DA could be responsible for the comparable temperature differences at the sensor positions, especially within the active segment of the DA. As shown in Figure 1a, the initial beam profile had two peaks along the active segment of the DA. However, the two peaks could fade at a far distance from the DA (a few mm; [29]) and become a single peak. In turn, the light interference at the far distance could cause the light intensity decrease in the ratio of 1/r^2^ (r = radial distance from the DA), resulting in less light absorption by the TC and eventually less measurement differences. Moreover, as the inner diameter of the esophagus is larger than 15 mm [3,6], both 18 and 22 mm OD glass tubes were comparatively tested (Figure 4), where each OD represented the distance between DA and W-TC. Compared to the 18 mm tube, the 22 mm tube gave a 32% lower measurement difference (ΔT_W-TC_ − ΔT_FBG_) possibly because of a larger radial distance of 2 mm (i.e., 3.5 ± 0.6 °C for 18 mm vs. 2.4 ± 0.3 °C for 22 mm), which also shows good agreement with previous studies [24,30,31,32]. Thus, the further distance of the TC from the DA can result in low artifacts (i.e., minimal light absorption by the TC) during the temperature measurements.

The surrounding medium (air or tissue) could influence the extent of the measurement differences between FBG and W-TC during laser irradiation. According to Figure 3a and Figure 4a, different temperature elevations (ΔT) were observed between air and tissue for the same temperature sensor (e.g., ΔT_W-TC_ = 6.3 ± 0.8 °C in air vs. 24.1 ± 3.1 °C in tissue; ΔT_FBG_ = 2.6 ± 0.3 °C in air vs. 21.4 ± 3.1 °C in tissue for M_0_ after 90-s irradiation). The current finding demonstrated a similar tendency of temperature elevations as in previous studies [33,34,35]. The profiles of the temperature developments (Figure 3 and Figure 4) follow the equation below:ΔT(t) = A(1 − e^−t/τ^) + Bt(1)
where A is the asymptotic temperature (°C), t is the time (s), τ is the time constant (s), A(1 − e^−t/τ^) is the temperature increase by light absorption of TC, and B (°C/s) is the slope of the temperature increase because of light absorption of a medium (air, water, or tissue) [35]. The amount of ΔT_W-TC_ can be determined by the direct light absorption of the TC in air (B = 0; Figure 3) and tissue (B ≠ 0; Figure 4). Upon laser irradiation, the initial tissue temperature measured by the TC increased rapidly as the incident laser energy was absorbed by the TC (Figure 3) or TC and tissue (Figure 4). Then, the temperature increase became steady due to thermal convection to the air or conduction to the surrounding tissue [36]. Therefore, the surrounding medium can determine the amount of light absorption and thus the extent of the temperature elevations for the two sensors. According to the ex vivo liver tests, the temporal developments of the temperature elevations (ΔT) were different between the glass tube and the balloon for both sensors (ΔT_W-TC_ = 13.7 ± 0.5 °C and ΔT_FBG_ = 12.2 ± 0.3 °C for the glass tube vs. ΔT_W-TC_ = 6.6 ± 0.2 °C and ΔT_FBG_ = 5.9 ± 0.4 °C for the balloon; Figure 4a and Figure 5a). Due to the lower thermal conductivity and larger thickness of the glass tube, the tube could have accompanied less conductive heat transfer to the surrounding media during irradiation, leading to a higher temperature increase.

As shown in Figure 6b, a significant difference in real-time temperature was observed between the inner (i.e., 28.3 ± 0.2 °C for FBG vs. 30.0 ± 0.1 °C for W-TC) and outer surfaces (i.e., 54.2 ± 3.6 °C for IR camera) of the esophagus tissue after 90-s laser irradiation. The current study used a cold filling medium to inflate a balloon as well as to cool down the mucosa temperature for tissue preservation during irradiation. In the treatment of the esophagus with gastroesophageal reflux disease, endoscopic laser irradiation can selectively increase the interstitial temperature in the muscle layer underneath the mucosa. The increase in temperature thereby induces coagulation of the muscle tissue without thermal injury to the mucosa [37], which can lead to muscle thickening, increased pressure in the low esophageal sphincter (LES), and the reduction of the nerve endings in the LES [38,39,40]. Figure 6c shows that the measurement difference (ΔT_W-TC_ − ΔT_FBG_) was 2.1 ± 0.4 °C. It was found that the total temperature elevations for both sensors were around 10 °C after the 90-s irradiation (28.2 ± 0.2 °C for FBG and 29.9 ± 0.1 °C for W-TC). In the case of preclinical and clinical models, the initial body temperature (36.2~37.5 °C [41]) is higher than that of the ex vivo tissue (20 °C). Although the overall temperature elevation would be larger than the current findings, various inherent factors, such as blood perfusion, dynamic tissue properties, and geometric conditions, would determine the final temperature. Therefore, further investigations should be performed in in vivo large animal models in order to confirm the measurement performance of the W-TC in comparison to the FBG for BCDA-assisted endoscopic laser treatment of the esophagus.

Although the current results demonstrated comparable performance between FBG and W-TC, experimental limitations remain. As ex vivo tissues (liver and esophagus) were used for quantitative comparison, temperature measurements hardly reflected the effect of blood perfusion (convective cooling), inherent and dynamic tissue properties (thermal and optical), non-uniformity of tissue components, initial body temperature, and partial or complete insulation in the tissue structure on measurement performance and accuracy. Despite minimal light absorption with a white tube, the tube covering the TC tip should be optimized in terms of tube colors, physical dimensions, and material properties. In addition, the BCDA-assisted laser application should be investigated under various tubular tissues (i.e., urethra, bile duct, duodenum) with different tissue properties (thermal and optical) to confirm the measurement performance of the W-TC. In the current study, a single wavelength (980 nm) in continuous mode was used to evaluate the two temperature sensors. In order to validate the performance of the W-TC, various wavelengths (400~2200 nm) should be tested with different light distributions of DAs and various irradiation times. Therefore, further studies will compare the performance of FBG and W-TC with various DAs in an in vivo porcine model to confirm the tendency and accuracy of the measurement difference between the two sensors and ultimately ensure the safety and efficacy of endoscopic laser application.

## 5. Conclusions

The current study compared the performance of FBG and W-TC during laser treatment of tubular tissue structures. The protection of the TC tip with a white tube could minimize light absorption and self-heating, as well as limit the measurement difference to less than 2 °C compared to the FBG. Further in vivo studies will be conducted to validate the current findings of the W-TC in terms of accuracy/precision of real-time temperature monitoring during endoscopic and circumferential laser irradiation for clinical translation.

## Figures and Tables

**Figure 1 sensors-23-06069-f001:**
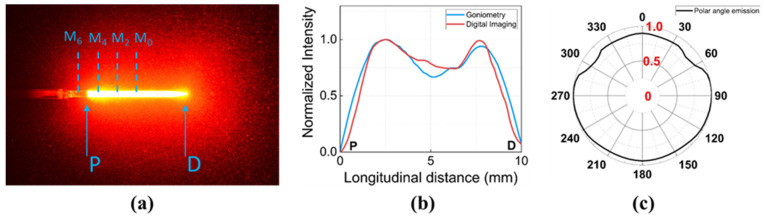
Characterization of diffusing optical fiber: (**a**) He-Ne light distribution along fiber, (**b**) longitudinal emissions from proximal (P) to distal (D) ends, and (**c**) polar emissions. Note that the sensor positions at 0, 2, 4, and 6 mm are indicated as M_0_, M_2_, M_4_, and M_6_, respectively.

**Figure 2 sensors-23-06069-f002:**
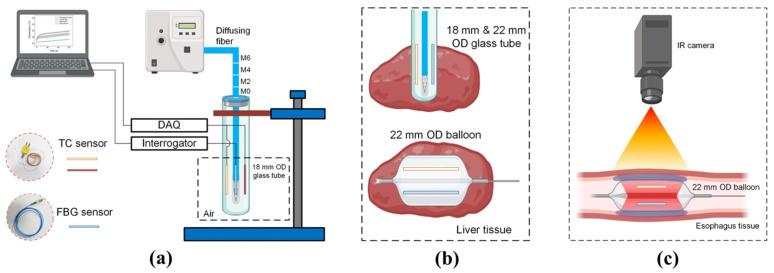
Schematic illustrations of experimental set-ups to evaluate temperature changes measured by Fiber Bragg Grating (FBG) and thermocouple (TC): (**a**) evaluations of white plastic tube covering TC tip, (**b**) ex vivo porcine liver tissue tests with 18- and 22-mm glass tubes (top) and balloon catheter-integrated diffusing applicator (bottom), and (**c**) ex vivo esophagus tissue tests. A 980 nm laser light was irradiated at 30 W for 90 s for all testing.

**Figure 3 sensors-23-06069-f003:**
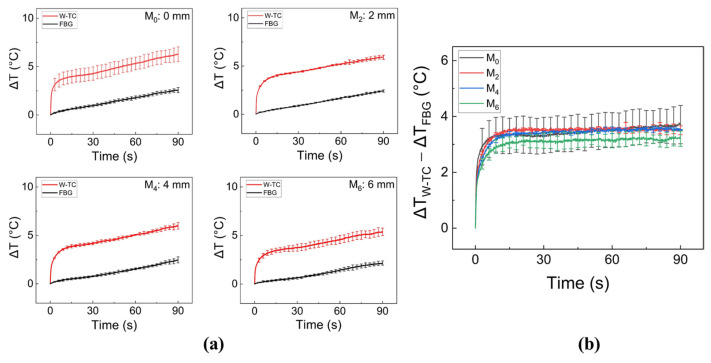
Comparison of temperature elevations on 18-mm glass tube measured at various positions (0, 2, 4, and 6 mm from the middle of the diffusing tip) by FBG and TC with white tubing cover (W-TC) during laser irradiation of 30 W for 90 s: (**a**) temperature elevations (ΔT) at various positions and (**b**) measurement differences between FBG and W-TC (ΔT_W-TC_ − ΔT_FBG_; N = 6). The sensor positions at 0, 2, 4, and 6 mm are denoted as M_0_, M_2_, M_4_, and M_6_, respectively.

**Figure 4 sensors-23-06069-f004:**
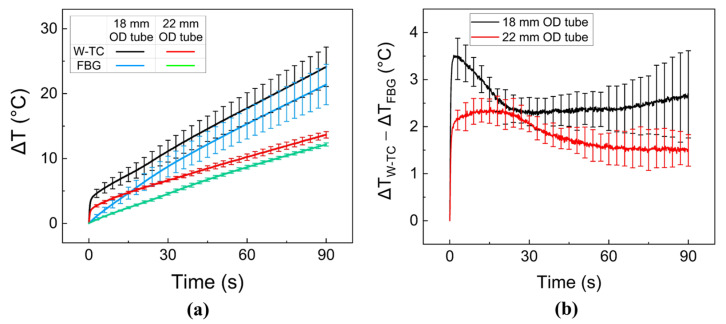
Comparison of temperature elevations at the tissue–glass tube interface using two different glass tubes (18 and 22 mm in outer diameter) on ex vivo porcine liver tissue during laser irradiation 30 W for 90 s: (**a**) temperature elevations measured by FBG and W-TC and (**b**) measurement differences between FBG and W-TC (ΔT_W-TC_ − ΔT_FBG_; N = 6). The sensor was positioned at the center of the diffusing tip (0 mm).

**Figure 5 sensors-23-06069-f005:**
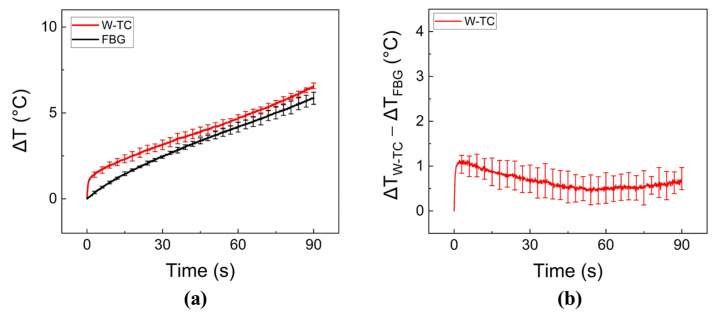
Evaluations of temperature elevations during laser irradiation of 30 W for 90 s on ex vivo porcine liver tissue using balloon catheter-integrated diffusing applicator: (**a**) comparison of temperature elevations at tissue-balloon interface measured by FBG and W-TC and (**b**) measurement differences between FBG and W-TC (ΔT_W-TC_ − ΔT_FBG_; N = 6).

**Figure 6 sensors-23-06069-f006:**
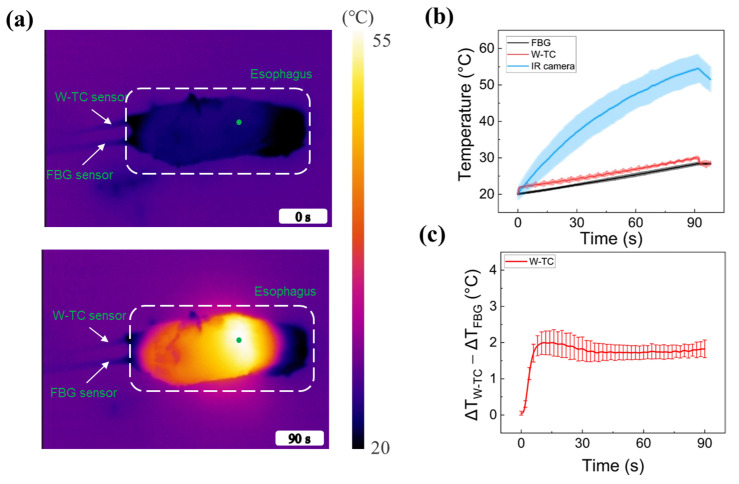
Thermal responses of ex vivo esophageal tissue to 980-nm laser irradiation (30 W for 90 s) using balloon catheter-integrated diffusing applicator: (**a**) infrared (IR) images captured before (top) and after (bottom) irradiation (N = 3), (**b**) comparison of temperatures in tissue lumen measured by FBG, W-TC, and IR camera (MP = measured point on outer surface), and (**c**) measurement differences between FBG and W-TC (ΔT_W-TC_ − ΔT_FBG_; N = 3).

## Data Availability

The data presented in this study are available on request from the corresponding author.

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
