# Peer review of "Comparative Evaluations on Real-Time Monitoring of Temperature Sensors during Endoscopic Laser Application"

_sensors, 2023, doi:10.3390/s23136069_

Round 1
Reviewer 1 Report
I recommend publication.
Reviewer 2 Report
The authors presents the "Comparative evaluations on real-time monitoring of temperature sensors during endoscopic laser application" overall study is promising, need a minor changes to accept:
I think line 52-72 should be modified according to the title of the manuscript, i think it should be related to temperature sensors and endoscopic laser application definition and problem statements.
Reviewer 3 Report
The author in this work has developed a temperature sensors, such as Fiber Bragg Grating (FBG) and thermocouple (TC), to monitor the interstitial tissue temperature during a laser irradiation during the endoscopic and circumferential laser treatment. Overall I'm impressed with the authors work, specially the way of their representation covering all the aspects. The work can be published after the authors address my few minor comments :
1. Did the authors also study the effect of laser wavelength and frequency during their study on temperature based sensor?
2. Can the authors comment how the conclusion of their study would get affected if they use other tissue samples than porcine esophageal?
